# A Multiwell-Plate *Caenorhabditis elegans* Assay for Assessing the Therapeutic Potential of Bacteriophages against Clinical Pathogens

Prasanth Manohar,[a,b] Belinda Loh,[a] Namasivayam Elangovan,[c] Archana Loganathan,[d] Ramesh Nachimuthu,[d] Sebastian Leptihn[a,e,f]

[a]Zhejiang University-University of Edinburgh (ZJE) Institute, Zhejiang University, School of Medicine, Haining, Zhejiang, People's Republic of China
[b]The Second Affiliated Hospital Zhejiang University (SAHZU), School of Medicine, Hangzhou, Zhejiang, People's Republic of China
[c]Department of Biotechnology, School of Bioscience, Periyar University, Salem, Tamil Nadu, India
[d]Antibiotic Resistance and Phage Therapy Lab, Department of Biomedical Science, School of Biosciences and Technology, Vellore, Tamil Nadu, India
[e]Department of Infectious Diseases, Sir Run Department Shaw Hospital, Zhejiang University, School of Medicine, Hangzhou, Zhejiang, People's Republic of China
[f]University of Edinburgh Medical School, Biomedical Sciences, College of Medicine & Veterinary Medicine, The University of Edinburgh, Edinburgh, United Kingdom

**ABSTRACT**  In order to establish phage therapy as a standard clinical treatment for bacterial infections, testing of every phage to ensure the suitability and safety of the biological compound is required. While some issues have been addressed over recent years, standard and easy-to-use animal models to test phages are still rare. Testing of phages in highly suitable mammalian models such as mice is subjected to strict ethical regulations, while insect larvae such as the *Galleria mellonella* model suffer from batch-to-batch variations and require manual operator skills to inject bacteria, resulting in unreliable experimental outcomes. A much simpler model is the nematode *Caenorhabditis elegans*, which feeds on bacteria, a fast growing and easy to handle organism that can be used in high-throughput screening. In this study, two clinical bacterial strains of *Escherichia coli*, one *Klebsiella pneumoniae*, and one *Enterobacter cloacae* strain were tested on the model system together with lytic bacteriophages that we isolated previously. We developed a liquid-based assay, in which the efficiency of phage treatment was evaluated using a scoring system based on microscopy and counting of the nematodes, allowing increasing statistical significance compared to other assays such as larvae or mice. Our work demonstrates the potential to use *Caenorhabditis elegans* to test the virulence of strains of *Klebsiella pneumoniae*, *Enterobacter cloacae*, and EHEC/EPEC as well as the efficacy of bacteriophages to treat or prevent infections, allowing a more reliable evaluation for the clinical therapeutic potential of lytic phages.

**IMPORTANCE** Validating the efficacy and safety of phages prior to clinical application is crucial to see phage therapy in practice. Current animal models include mice and insect larvae, which pose ethical or technical challenges. This study examined the use of the nematode model organism *C. elegans* as a quick, reliable, and simple alternative for testing phages. The data show that all the four tested bacteriophages can eliminate bacterial pathogens and protect the nematode from infections. Survival rates of the nematodes increased from <20% in the infection group to >90% in the phage treatment group. Even the nematodes with poly-microbial infections recovered during phage cocktail treatment. The use of *C. elegans* as a simple whole-animal infection model is a rapid and robust way to study the efficacy of phages before testing them on more complex model animals such as mice.

**KEYWORDS** phage therapy, animal infection model, *Caenorhabditis elegans*, phage efficacy, bacterial pathogens, bacteriophage therapy, bacteriophages

Address correspondence to Ramesh Nachimuthu, drpnramesh@gmail.com, or Sebastian Leptihn, Leptihn@intl.zju.edu.cn.

The authors declare no conflict of interest.

Gram-negative bacteria can cause serious infections in humans, including strains from the family of the *Enterobacteriaceae* (*Escherichia coli*, *Klebsiella pneumoniae*, and *Enterobacter cloacae*). Over the years, such pathogenic strains have been exhibiting increasing resistance to last-resort antibiotics such as carbapenems and colistin (1–3). With the decreasing number of available antibiotics, global overuse, and misuse in both clinical medicine and agriculture, antimicrobial resistance is an undeniable reality that is also unavoidable (4). However, with appropriate governmental regulations concurrent with the education of doctors and patients alike, the velocity at which resistant strains emerge may be decreased. Nonetheless, novel antimicrobial compounds that can be used in medicine and/or agriculture would not only benefit the sectors but are essential for global health. In light of the withdrawal of pharmaceutical global players to discover new classes of small molecule antibiotics, an alternative approach is to use bacteriophages as biological therapeutics (5, 6).

Phage therapy is the application of bacterial viruses that kill their hosts and thus eliminate a bacterial infection. Although phage therapy is an old concept, it is necessary to improve the understanding of antibacterial properties of phages and their applications before bringing phages into clinical practice (7). Bacteriophages are known to be genus-, species-, or even strain-specific and exhibit no or only minimal side effects when infecting pathogens inside a mammalian host (8–10). Their interactions within mammalian hosts, in particular with the immune system, are complex, and a healthy phageome has been shown to be beneficial to the host (11–14). In order to translate a phage isolate into a "drug," several challenges need to be addressed, such as evaluating host susceptibility to the phage to ensure that the phage can infect and kill the pathogen, genomic analysis to guarantee the absence of virulence factors or lysogeny genes, producing phages at high numbers devoid of any toxic compounds such as bacterial LPS, and testing the efficacy of the phage in an animal model (15, 16).

There is a need to improve the current workflow, from phage discovery to the application of phages, which includes investigating the efficacy of a phage in a simple animal model (17, 18). Many studies have reported the use of wax worms (*Galleria mellonella*) as a model to study the efficacy of bacteriophages (15, 19–22). However, it requires a skillful operator to inject bacteria and/or phages into individual larvae, which sometimes could result in large variations between batches and thus lead to nonreproducible observations. While ethical approval is easier to obtain for work with *G. mellonella*, large numbers of larvae are needed to achieve statistical relevance and therefore a considerable amount of time is required to reliably test a phage. While a mammalian host such as mice would be most suitable to test a phage prior to clinical use, ethical approval is difficult to obtain and experiments take too long (especially for emergency use of phages in compassionate therapy), while low animal numbers also decrease statistical significance. Thus, a robust, reliable, and reproducible assay with large numbers of animals would be highly advantageous to test the efficacy of a phage.

In this study, we used the nematode *Caenorhabditis elegans* as a simple whole-animal infection model and established a liquid-based assay in a multiwell format. *C. elegans* is an established animal infection model to study pathogenesis and to evaluate the efficacy of drugs (23–27). However, and perhaps surprisingly, there are very limited studies that report the use of *C. elegans* as an animal model to evaluate the efficacy of phage therapy (28, 29). Studies have been conducted to investigate virulence mechanisms of *Pseudomonas aeruginosa* (30); *Burkholderia pseudomallei* (31); and *Salmonella typhimurium*, *S. enteritidis*, and *S. pullorum* (29, 32, 33). In addition, infections of *C. elegans* by pathogenic *E. coli*, *K. pneumoniae*, *E. cloacae*, and the Gram-positive *Staphylococcus aureus* have been studied where the mechanisms of host–pathogen interaction were elucidated, but also the antibacterial activity of phages were examined (23, 28). In all of these studies, solid media were used in the experiments, making it a complex, and possibly suboptimal, assay. The goal of our study was to establish a *C. elegans*-pathogen (*E. coli*, *K. pneumoniae*, and *E. cloacae*) liquid-based platform to elucidate the efficacy of lytic phages *in vivo*. Here, we were able to demonstrate that in the presence of lytic phages the life span of infected nematodes was increased up to 6-fold compared to the controls without phage. In the bacteria infected groups, the nematode survival

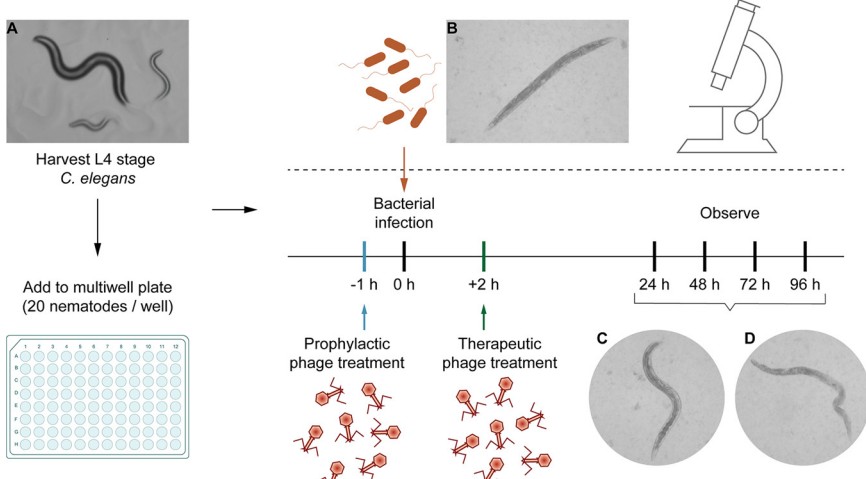

**FIG 1** Schematic representation of the assay and distinct appearances of the nematode *C. elegans* on solid or in liquid medium. Representative microscopic images may serve as a guide for scoring by the operator. Movement is an important indicator for the viability of a worm but not a strict requirement to score a nematode to be alive. (A) *C. elegans* on NGA plates. (B, C) Live L4 stage mature nematodes in liquid medium. (D) Dead nematode after bacterial infection. The worms were examined under an inverted microscope at $\times 40$ magnification.

was reduced by ∼80% within 4 days. Interestingly, the prophylactic treatment (phages introduced 1 h before bacterial infection) showed better efficacy than the therapeutic treatment group (phages introduced 2 h after bacterial infection) across all four tested pathogens.

## RESULTS

***C. elegans* as infection models.** We established a *C. elegans* liquid-based assay with the aim of providing a robust and reliable approach with high numbers of repeats for statistical significance, to study the virulence of bacterial isolates *in vivo* and the effects of bacteriophages that target these pathogens. Prior to experimental conditions, we conducted a series of control experiments. In our first control, *C. elegans* were fed with the nonpathogenic *E. coli* strain OP50 and were observed to live up to 5 days. In the case where no bacteria were added as feed, the life span of *C. elegans* was not drastically reduced despite the starvation conditions (Fig. S1 in the supplemental material).

It had previously been shown that several pathogens have the ability to kill *C. elegans* including *Pseudomonas* and *Klebsiella* when the nematodes were exposed to the bacteria in the surrounding media. Thus, nematodes can be used as a model organism to study bacterial infection (34, 35). In addition, despite being the bacterium that *C. elegans* can feed on, studies have shown that certain *E. coli* strains that are pathogenic in humans or animals have the ability to infect nematodes as well (36). Therefore, in our second set of control experiments, we sought to determine the pathogenicity of bacterial strains on *C. elegans*. Groups of 20 nematodes were independently infected with 4 pathogenic strains, *Escherichia coli* 131, *Escherichia coli* 311, *Klebsiella pneumoniae* 235, and *Enterobacter cloacae* 140, and observed over 5 days. Regardless of strain, when the nematodes were infected with $10^3$ CFU/mL of bacterial pathogen, they lived beyond 5 days, whereas $10^7$ CFU/mL caused death within 3 days (Fig. S2).

As our experimental setup was conducted in 96-well plates, the maximum volume in each well was 100 $\mu$L. Over extended durations lasting longer than 5 days, the experimental conditions would lead to volume reduction before eventually drying up completely, thus altering the conditions and potentially affecting the results of our study. Therefore, in this study, a bacterial concentration of $10^5$ CFU/mL was chosen as all nematodes succumbed to infection in less than 5 days.

Nematodes were evaluated based on appearance (reflectance and optical transparency) and mobility (alive: moving; dead: inactive) as observed under an inverted microscope ($\times 40$ magnification). A representative scoring model is provided in Fig. 1. Scoring was performed

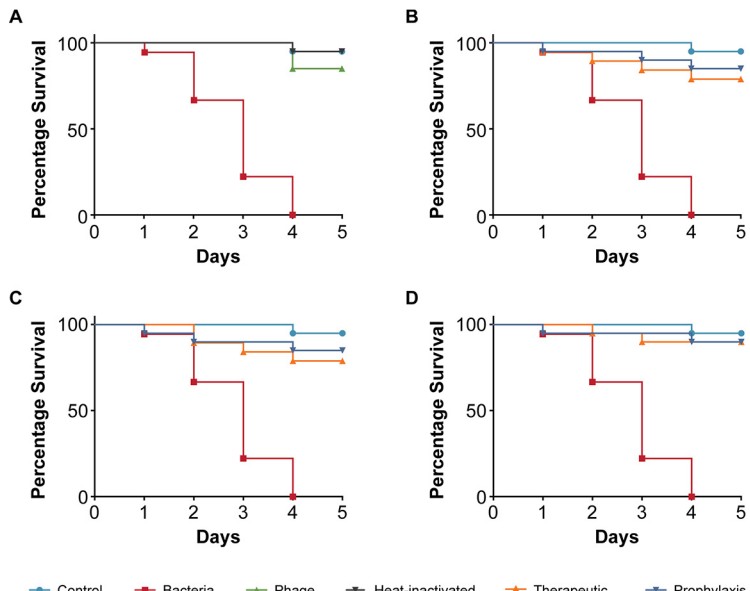

**FIG 2** Pathogenicity of *Escherichia coli* strain 131 in *C. elegans* and efficacy of *Escherichia* phage myPSH1131 against *E. coli* infections. The control group consisted of *C. elegans* fed with *E. coli* OP50 and exposed to *E. coli* strain 131 ($OD_{600}$ = 0.6) that kills *C. elegans* in liquid medium. Twenty nematodes were used in each group. Representative survival curves of *C. elegans* following infection by *E. coli* strain 131 in (A) liquid medium consisting of M9 buffer and *E. coli* culture, *Escherichia* phage, or heat-inactivated bacteria and (B, C, D) survival curves of *C. elegans* following infection with *E. coli* strain 131 and treatment with *Escherichia* phage, therapeutic and prophylactic treatment. (B) Survival curves of *C. elegans* infected and treated with bacteria and phage ratio of 1:1, i.e., $10^5$ CFU/ mL and $10^5$ PFU/mL. (C) Survival curves of *C. elegans* infected and treated with bacteria and phage ratio of 1:10, i.e., $10^5$ CFU/mL and $10^6$ PFU/mL. (D) Survival curves of *C. elegans* infected and treated with bacteria and phage ratio of 1:100, i.e., $10^5$ CFU/mL and $10^7$ PFU/mL. The survival curves were plotted using the Kaplan-Meier method, and the log-rank test was used to analyze the difference in survival rates in GraphPad Prism 7.0. A statistically significant difference ($P < 0.05$) was observed in the treatment groups.

in 24 h intervals. We classified phages to be effective if nematode survival was higher than 70% of the infected nematodes, while less than 20% survival in the infected groups was classified as ineffective.

In order to determine if our model organism is sensitive to endotoxin, four live or heat-inactivated pathogenic bacterial strains (*E. coli* 131, *E. coli* 311, *K. pneumoniae* 235, and *E. cloacae* 140) were tested on the nematodes. When the *C. elegans* were infected with live pathogenic *E. coli*, *K. pneumoniae*, or *E. cloacae* strain, a significant reduction in nematode survival was observed. In contrast, nematode survival was maintained at 95% when infected with heat-inactivated bacteria, indicating that heat stable molecules such as the endotoxin LPS do not lead to mortality of the worms (Fig. S3). Thus, our experiments show that it is indeed the pathogenicity of the bacteria used in this study that kills the nematode.

**Phage therapy enhanced the survival of infected nematodes.** The first isolate we tested in our study was *E. coli* 131, a clinical strain from a diagnostic center in Chennai (India), isolated from blood and previously identified to be enteropathogenic (an EPEC strain) by PCR (37, 38). We established the parameter of $TD_{mean}$, which we defined as the mean time until we observed the nematodes die. The $TD_{mean}$ for *E. coli* strain 131-infected worms was found to be 3 days, a significant reduction from the usual life span of 5 days (Fig. 2A). *Escherichia* phage myPSH1131 was previously isolated and identified to infect and lyse *E. coli* strain 131 effectively (37, 38). When myPSH1131 was tested alone, no adverse effects on *C. elegans* were observed (Fig. 2A). This shows either that the phage preparations were free of toxic substances that could decrease the *C. elegans'* life span or that the nematode is not affected by any of the components contained in the solution. In order to study the efficacy of phage treatment, varying ratios of bacteria to phage were tested, i.e., 1:1, 1:10, and 1:100, which were $10^5$ CFU/mL:$10^5$ PFU/mL, $10^5$ CFU/mL:$10^6$ PFU/mL, and $10^5$

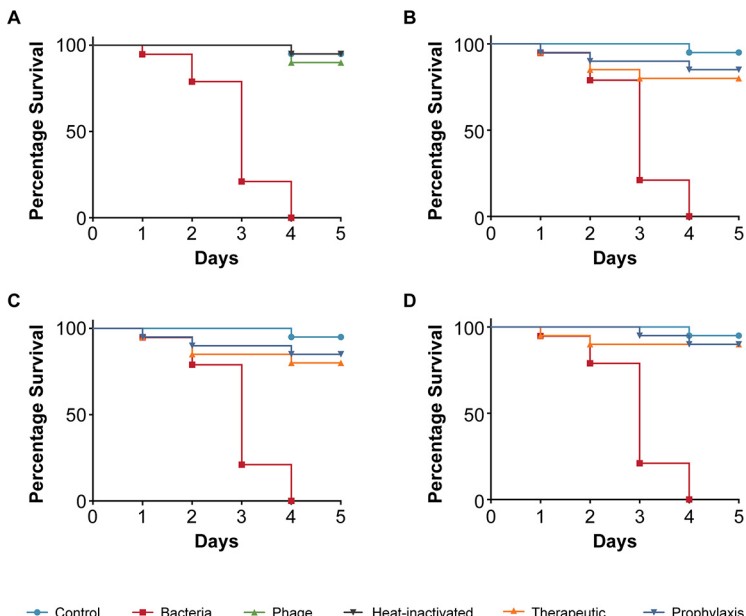

**FIG 3** Pathogenicity of *Escherichia coli* strain 311 in *C. elegans* and efficacy of *Escherichia* phage myPSH2311 against *E. coli* infections. The control group consisted of *C. elegans* feed with *E. coli* OP50 and *E. coli* strain 311 ($OD_{600}$ = 0.6) that kills *C. elegans* in liquid medium. Twenty nematodes were used in each group. Representative survival curves of *C. elegans* following infection by *E. coli* strain 311 in (A) liquid medium consisting of M9 buffer and *E. coli* culture, *Escherichia* phage, heat-inactivated bacteria and (B, C, D) survival curves of *C. elegans* following infection with *E. coli* strain 311 and treatment with *Escherichia* phage, therapeutic and prophylactic treatment. (B) Survival curves of *C. elegans* infected and treated with bacteria and phage ratio of 1:1, i.e., $10^5$ CFU/mL and $10^5$ PFU/mL. (C) Survival curves of *C. elegans* infected and treated with bacteria and phage ratio of 1:10, i.e., $10^5$ CFU/mL and $10^6$ PFU/mL. (D) Survival curves of *C. elegans* infected and treated with bacteria and phage ratio of 1:100, i.e., $10^5$ CFU/mL and $10^7$ PFU/mL. The survival curves were plotted using the Kaplan-Meier method, and the log-rank test was used to analyze the difference in survival rates in GraphPad Prism 7.0. A statistically significant difference ($P < 0.05$) was observed in the treatment groups.

CFU/mL:$10^7$ PFU/mL. In the therapeutic treatment group, phage was added after exposure of the nematode to the pathogen. Here, the *Escherichia* phage myPSH1131 was able to increase survival up to 90% when the bacteria to phage ratio was 1:100, i.e., $10^5$ CFU/mL against $10^7$ PFU/mL (Fig. 2D). Other ratios tested, i.e., 1:1 and 1:10, also showed survival up to 5 days. (Fig. 2B and C). Similar results were observed for the prophylactic treatment group, with nematode survival up to 90% after 5 days in 1:100 (Fig. 2D). In this group, the phage was added 1 h before the pathogen was included into the media.

Next, we tested the *E. coli* strain 311 that was isolated from blood and found to be enterohemorrhagic, i.e., an EHEC strain as identified by PCR to classify pathotypes. Here, the $TD_{mean}$ in infected *C. elegans* was 3 days and no toxicity was observed when the strain specific phage myPSH2311 was used alone (Fig. 3A). *E. coli* strain 311-infected nematodes treated with the *Escherichia* phage resulted in 90% survival after 5 days in both therapeutic and prophylactic groups (Fig. 3D). At lower phage concentrations, however, 1:1 and 1:10, the survival was up to 80% after 4 days (Fig. 3B and C).

Next, we tested *K. pneumoniae* strain 235, which was isolated from the blood of a patient. Infection with *K. pneumoniae* 235 was significantly more virulent to *C. elegans* compared to the two *E. coli* strains we had tested. The infection killed all the nematodes within 4 days, and the $TD_{mean}$ was established to be less than 3 days (Fig. 4A). The addition of phage only had no adverse effects on nematode survival. When phages were added to nematodes that had been exposed to *K. pneumoniae* strain 235, i.e., the treatment group, nematode survival increased to 80% irrespective of the phage concentrations that we analyzed and the worms stayed alive at 5 days (Fig. 4C and D). The prophylactic treatment had a slightly increased positive impact on nematode survival when the bacteria to phage ratio was 1:100, with 90% survival after 5 days, compared to the therapeutic group (Fig. 4D).

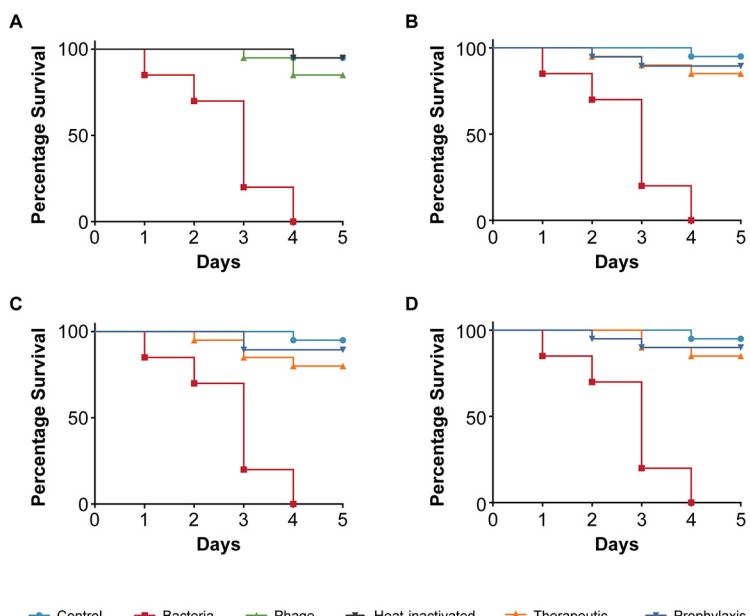

**FIG 4** Pathogenicity of *Klebsiella pneumoniae* strain 235 in *C. elegans* and efficacy of *Klebsiella* phage myPSH1235 against *K. pneumoniae* infections. The control group consisted of *C. elegans* fed with *E. coli* OP50 and exposed to *K. pneumoniae* strain 235 ($OD_{600}$ = 0.6) that kills *C. elegans* in liquid medium. Twenty nematodes were used in each group. Representative survival curves of *C. elegans* following infection by *K. pneumoniae* strain 235 in (A) liquid medium consisting of M9 buffer and *K. pneumoniae* culture, *Klebsiella* phage, or heat-inactivated bacteria and (B, C, D) survival curves of *C. elegans* following infection with *K. pneumoniae* strain 235 and treatment with *Klebsiella* phage, therapeutic and prophylactic treatment. (B) Survival curves of *C. elegans* infected and treated with bacteria and phage ratio of 1:1, i.e., $10^5$ CFU/mL and $10^5$ PFU/mL. (C) Survival curves of *C. elegans* infected and treated with bacteria and phage ratio of 1:10, i.e., $10^5$ CFU/mL and $10^6$ PFU/mL. (D) Survival curves of *C. elegans* infected and treated with bacteria and phage ratio of 1:100, i.e., $10^5$ CFU/mL and $10^7$ PFU/mL. The survival curves were plotted using the Kaplan-Meier method, and the log-rank test was used to analyze the difference in survival rates in GraphPad Prism 7.0. A statistically significant difference ($P < 0.05$) was observed in the treatment groups.

As a fourth pathogenic strain, we tested an *Enterobacter cloacae* isolate although *Enterobacter* is known to often be part of the commensal flora in *C. elegans* (39). However, no *Enterobacter* species were obtained in our *C. elegans* when attempting to specifically isolate such species. Similarly to the pathogenic *E. coli* strain 140 compared to the *E. coli* feeding strain OP50, the observations we made illustrate that strains of the same species can exhibit fundamentally different effects on the host, likely due to virulence factors. The *E. cloacae* strain 140 is a clinical isolate (from a urine sample) obtained from a diagnostic center in Chennai (India), which resulted in mortality of infected nematodes with a $TD_{mean}$ of 3 days (Fig. 5A). While purified *Enterobacter* phage myPSH1140 alone had no negative effect on the nematodes (survival >90%), the therapeutic treatment group showed increased survival to 85% at 5 days when 1:100 was used (Fig. 5D). However, both 1:1 and 1:10 had survival up to 75% at 5 days (Fig. 5B and C). The prophylactic phage treatment allowed survival of 85% of the nematodes when 1:100 was used but up to 80% survival when 1:1 or 1:10 was used at the same time point (Fig. 5B to D).

Virulence is often assessed with a single pathogen. However, many infections in clinical practice are not caused by a single pathogen but by several, sometimes with opportunistic bacteria complicating the infection. Such poly-microbial infections are rarely investigated in the lab. Here we used a mixture of the pathogens (*E. coli* 131, *E. coli* 311, *K. pneumoniae* 235, *E. cloacae*), which we had tested individually and investigated in our animal model. Such poly-microbial infections significantly reduced the survival percentage to 50% within 2 days, possibly displaying synergistic virulence effects, not cumulative ones as we used the same final number of cells (Fig. 6A). When we next investigated if phages had a beneficial effect decreasing mortality of the nematodes, we prepared a so-called "phage cocktail" that

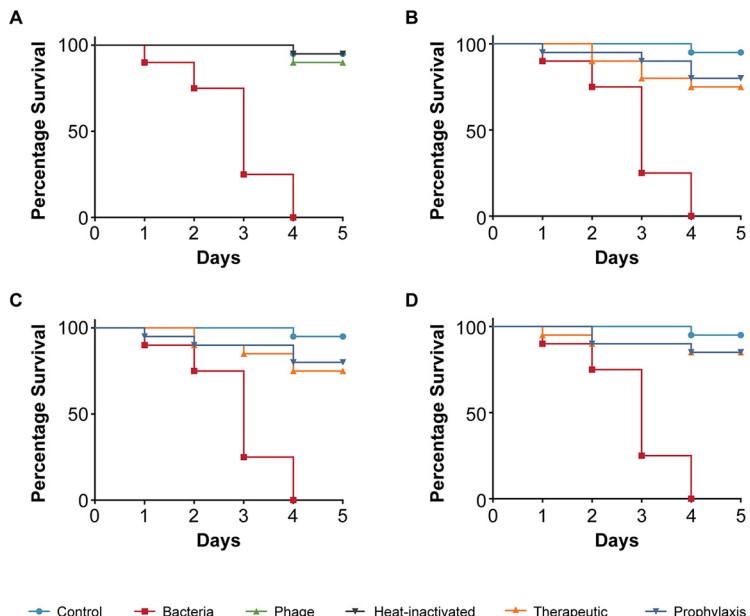

**FIG 5** Pathogenicity of *Enterobacter cloacae* strain 140 in *C. elegans* and efficacy of *Enterobacter* phage myPSH1140 against *E. cloacae* infections. The control group consisted of *C. elegans* fed with *E. coli* OP50 and exposed to *E. cloacae* strain 140 ($OD_{600}$ = 0.6) that kills *C. elegans* in liquid medium. Twenty nematodes were used in each group. Representative survival curves of *C. elegans* following infection by *E. cloacae* strain 140 in (A) liquid medium consisting of M9 buffer and *E. cloacae* culture, *Enterobacter* phage, heat-inactivated bacteria and (B, C, D) survival curves of *C. elegans* following infection with *E. cloacae* strain 140 and treatment with *Enterobacter* phage, therapeutic and prophylactic treatment. (B) Survival curves of *C. elegans* infected and treated with bacteria and phage ratio of 1:1, i.e., $10^5$ CFU/mL and $10^5$ PFU/mL. (C) Survival curves of *C. elegans* infected and treated with bacteria and phage ratio of 1:10, i.e., $10^5$ CFU/mL and $10^6$ PFU/mL. (D) Survival curves of *C. elegans* infected and treated with bacteria and phage ratio of 1:100, i.e., $10^5$ CFU/mL and $10^7$ PFU/mL. The survival curves were plotted using the Kaplan-Meier method, and the log-rank test was used to analyze the difference in survival rates in GraphPad Prism 7.0. A statistically significant difference ($P < 0.05$) was observed in the treatment groups.

contained phages infecting all the pathogens we used in our poly-microbial infection test. When investigating the effect of the phage cocktail alone on the nematodes, no negative influence on survival was observed. Using the mixture of phages in worms exposed to the pathogens, the survivability of nematodes was up to 75% (therapeutic) and 80% (prophylaxis) when the phage concentration was higher, i.e., 1:100 (Fig. 6D). A reduction in survival was observed when the phage concentration against bacteria was reduced (Fig. 6B and C). This observation demonstrates the efficacy of phage cocktails in reducing the bacterial load caused by the tested poly-microbial infections (Fig. 6B to D).

As phages can infect and inactivate bacteria in the media surrounding the worm, we next investigated if the phages are being taken up by the nematodes. The amount of phage particles inside the worms was reduced to $10^2$ PFU/mL from an initial concentration of $10^6$ PFU/mL after 4 days in a solution containing only the phage (Fig. 7). However, a 10-fold higher concentration of phages ($\sim 10^3$ PFU/mL) was found in the therapeutic and in the prophylactic treatment groups, indicating that phage replication occurs inside the nematode (Fig. 7).

For all four bacteriophages, the phage treatment resulted in a 6-fold or higher increase in nematode survival. In all cases, the prophylactic treatment resulted in a higher percentage of nematode survival compared to the therapeutic treatment. This is possibly due to a decrease in bacterial load as phages are able to kill bacteria that are not yet being ingested by the nematode. Regardless of this issue, which is not straightforward to test, our work has clearly established that *C. elegans* can serve as a robust, reproducible, statistically valuable testing platform for assessing not only the virulence of bacterial pathogens but more importantly the efficacy of potentially therapeutic phages, a first step toward ensuring safe deployment in phage therapy.

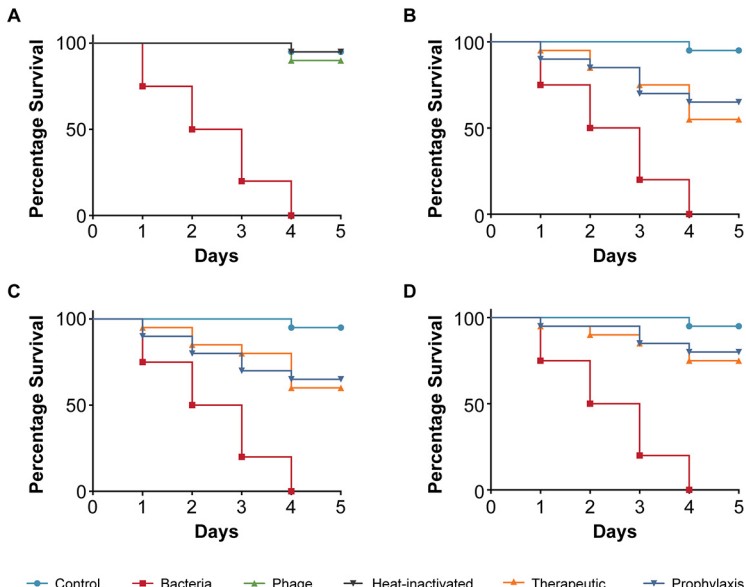

**FIG 6** Pathogenicity of multibacterial culture in *C. elegans* and efficacy of phage cocktails against poly-microbial infections. The control group consisted of *C. elegans* fed with *E. coli* OP50. Twenty nematodes were used in each group. Poly-microbial culture (*E. coli* 131, *E. coli* 311, *K. pneumoniae* 235, *E. cloacae* 140) kills *C. elegans* in liquid medium. Representative survival curves of *C. elegans* following infection by poly-microbial culture in (A) liquid medium consisting of M9 buffer and bacterial culture or phage cocktail or head-inactivated bacteria and (B, C, D) survival curves of *C. elegans* following infection with poly-microbial bacteria and treatment with phage cocktail, therapeutic and prophylactic treatment. (B) Survival curves of *C. elegans* infected and treated with poly-microbial bacteria and phage cocktail ratio of 1:1, i.e., $10^5$ CFU/mL and $10^5$ PFU/mL. (C) Survival curves of *C. elegans* infected and treated with poly-microbial bacteria and phage cocktail ratio of 1:10, i.e., $10^5$ CFU/mL and $10^6$ PFU/mL. (D) Survival curves of *C. elegans* infected and treated with poly-microbial bacteria and phage cocktail ratio of 1:100, i.e., $10^5$ CFU/ mL and $10^7$ PFU/mL. Survival curves were plotted using the Kaplan-Meier method, and the log-rank test was used to analyze the difference in survival rates in GraphPad Prism 7.0. A statistically significant difference ($P < 0.05$) was observed in the treatment groups.

## DISCUSSION

We have successfully established a liquid-based *C. elegans* screening platform to investigate the efficacy of potential therapeutic bacteriophages. This robust and reproducible approach to study the antibacterial potential of lytic phages using a whole-animal infection model is valuable not only because of its ease but also due to better statistics with higher numbers of animals compared to other established assays. We were able to show that our method appears to be employable for a diverse range of pathogens that cause diseases in humans, as they also cause infections in *C. elegans* (40). Most studies involving *C. elegans* are routinely performed on agar plates. However, the use of a liquid-based screening method has several advantages including time efficiency and easy experimental design, as components can be added to the solution, which allows for an even exposure of the nematodes to bacteria and/or bacteriophage. The screening on titer plates allows quick nematode scoring, i.e., makes easy and clear distinguishing of nematodes, as the colorless M9 buffer allows the nematodes to be observed clearly in contrast to solid media. Even though some experience is required to conduct worm survival counts, the skill can be acquired easily. A liquid-based screening with *C. elegans* was previously found to be suitable to address the virulence of *Staphylococcus aureus* (26). The method employs an approach similar to ours; using *C. elegans* as a model organism in liquid media requires high-end equipment, and the use of bacteriophages has not been established with this method (27). In addition, our assay is easier to use, faster, and less cost-intensive. However, in its current form, our assay can be regarded as semiquantitative as a human operator evaluates the viability of the nematodes. Thus, the development of a fully quantitative assay would be an improvement for future optimization of the method. Potential approaches might include a robotic, automated counting of worms using, e.g., specifically developed

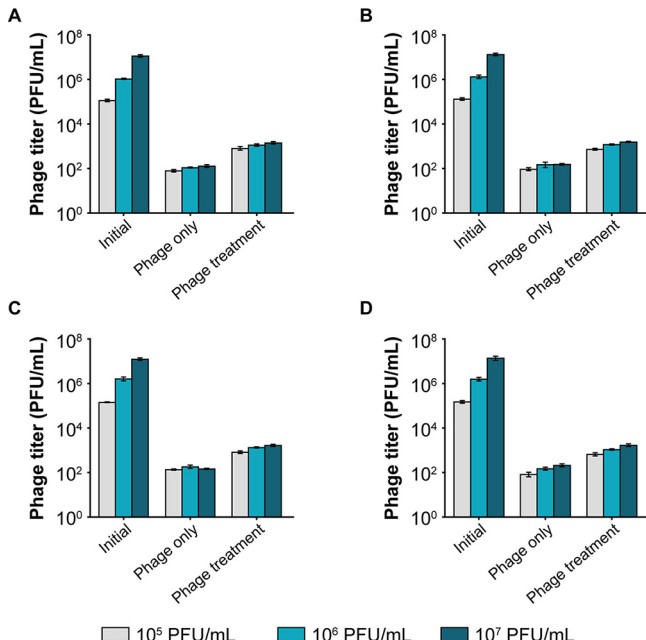

**FIG 7** Bacteriophage enumeration from *C. elegans* after 4 days of exposure under two different conditions: (i) phage only, i.e., *C. elegans* was introduced to phage alone, and (ii) phage treatment i.e., *C. elegans* infected with bacteria was treated with phages. (A) *Escherichia* phage myPSH1131. (B) *Escherichia* phage myPSH2311. (C) *Klebsiella* phage myPSH1235 and (D) *Enterobacter* phage myPSH1140. After 4 days of treating *C. elegans* with phages, the nematodes were washed, ground and titrated for the presence of phages. The number of phages in the treatment group was 10 times higher than the untreated group. The error bars represent the standard mean of three independent experiments.

image processing software (37), or an antibody-based ELISA against live/dead specific marker proteins.

Using the simple animal model in our assay, we have also demonstrated the efficacy of four bacteriophages in increasing the survival of *C. elegans* infected with bacterial pathogens. The four bacteriophages (*Escherichia* phage myPSH1131, *Escherichia* phage myPSH2311, *Klebsiella* phage myPSH1235, and *Enterobacter* phage myPSH1140) were found to be effective in eliminating the bacterial load, increasing the life span of *C. elegans* (Fig. 2 and 5). Although the higher concentration of bacteria–phage ratio (1:100 or $10^5$ CFU/mL:$10^7$ PFU/mL) showed up to 90% nematode survival, the lesser concentrations of 1:1 and 1:10 had at least 80% survival. Therefore, survivability of the nematodes was observed irrespective of the phage concentration. Also in combination, as a phage cocktail, the phages displayed highly beneficial effects in limiting the impact of poly-microbial infections on the nematodes. To the best of our knowledge, this is the first study to report the successful use of *C. elegans* to test the efficacy of phages in a liquid assay format.

The data from our assay confirms results we have previously obtained in experiments using wax worms (15). In our previous study, when wax worms were infected with bacteria ($10^8$ CFU/mL) and treated with phages ($10^4$ PFU/mL), up to 80% larval survival was noted (15). In this study, perhaps not surprisingly, the prophylactic treatment allows for better nematode survival rates than the therapeutic treatment where an infection is first established before treating the worm with phages. However, even in the treatment regime we obtained statistically significant high survival rates, which demonstrate that bacteriophages are able to protect or cure *C. elegans* from bacterial infections caused by the four different clinical strains, also when combined in an effort to reproduce a ploy-microbial infection. We believe that this is the first study to test the pathogenesis of poly-microbial infections in *C. elegans* and also to use phage cocktails against the poly-microbial infections.

As phage therapy becomes an increasingly used clinical intervention to treat MDR infections, the use of simple live animal models in robust and reliable assays like the one we have established for *C. elegans* will facilitate the therapeutic evaluation of phages. While

innate immune factors are conserved among nematodes, insects, and mammals, there are distinct differences among lower animals (38, 41). Both, *Galleria* and *Caenorhabditis* lack an adaptive immune system; however, studies on the innate immune systems of insects and nematodes have indicated that there are differences in pattern-recognition molecules and signaling factors. Although results of our work do not indicate any difference (such as a higher/lower tolerance to certain pathogens in *C. elegans*), the insect and the nematode models might show variances when evaluating the virulence of bacteria. However, for clinical phage therapy, in particular for human applications, an animal model with an adaptive immune system should be used. Thus, an invertebrate-based screening can only serve as the first stage to evaluate the therapeutic potential of a phage, yet in our case as a rapid and inexpensive one. Such robust, reliable, and potentially high-throughput methods can be considered one of the prerequisites for the implementation of phage therapy, from the discovery of environmental phages or the construction of synthetic viruses, to the phage solution ready for medical treatment.

The data provided in this study demonstrate the ability of a liquid-based assay to test the antimicrobial efficacy of bacteriophages to increase the nematode survival when infected with bacterial pathogens. This allows the activity of phages to be tested before large-scale preclinical studies in mouse models. While the use of *C. elegans* was explored in this study by manual operation, observation, and analysis, modifications to our test system could allow the adaptation to establish a high-throughput screening platform for hundreds of bacteriophages in parallel.

## MATERIALS AND METHODS

**Bacterial strains and bacteriophages.** A total of four clinical bacterial strains, *Escherichia coli* 131 (enteropathogenic), *Escherichia coli* 311 (enterohemorrhagic), *Klebsiella pneumoniae* 235, and *Enterobacter cloacae* 140, were previously studied and chosen for the study (42, 43). *E. coli* pathotypes were identified by PCR as detailed in our previous studies (42, 43). Bacterial strains including the nonpathogenic "nematode feeding bacterium" *E. coli* OP50 were grown on Lysogeny broth (LB) agar at 37°C and stored in LB medium. For the assay, bacterial strains were grown overnight at 37°C with shaking at 120 rpm, then the culture was diluted to $OD_{600} = 0.6$ ($\sim 10^5$ CFU/mL) unless stated otherwise. Heat-inactivated bacteria were prepared by incubating the cells at 65°C for 40 min (44). One mL of cells was pelleted by centrifugation at $4,000 \times g$ for 10 min before washing the pellet thrice with phosphate-buffered saline (PBS) and finally resuspending it in 1 mL PBS. Four bacteriophages, *Escherichia* phage myPSH1131, *Escherichia* phage myPSH2311, *Klebsiella* phage myPSH1235, and *Enterobacter* phage myPSH1140 were used in this study (42, 43). The purified bacteriophages were prepared at $10^5$–$10^7$ PFU/mL as described previously (42, 43). All the bacterial strains and bacteriophages used in this study are available at Antibiotic Resistance and Phage Therapy Laboratory, Vellore Institute of Technology, Vellore, India.

***C. elegans* maintenance.** Bristol N2 (wild-type) *C. elegans* was used in this study and was kindly provided by Dr. N. Elangovan, Periyar University, Salem, India. The nematodes were maintained and propagated on nematode growth media (17 g agar, 3 g NaCl, 2.5 g peptone, 0.5 mL of 1M $CaCl_2$, 1 mL of 5 mg/mL cholesterol, 1 mL of 1M $MgSO_4$, 25 mL $KH_2PO_4$ buffer [pH 6.0] per L) plates that carry *E. coli* OP50 as a source of food at 20°C by standard protocols (45). The adult worms were exposed to 5 M sodium hydroxide and 5% bleach to collect eggs that were then incubated in M9 medium (6 g $Na_2HPO_4$, 3 g $KH_2PO_4$, 5 g NaCl, 0.25 g $MgSO_4.7H_2O$). Worms take approximately 12 h to hatch, and only mature eggs allow the development of viable nematodes by which synchronization was archived to obtain worms of the same age (46). For the experiments, age synchronized L4 worms were incubated in the presence of tryptic soy broth (TSB), *E. coli*, *K. pneumoniae*, *E. cloacae* and/or bacteriophage of *Escherichia* phage myPSH1131, *Escherichia* phage myPSH2311, *Klebsiella* phage myPSH1235, and *Enterobacter* phage myPSH1140 in 96-well microtiter plates containing 100 $\mu$L of M9 buffer in each well.

**Testing phage efficacy in the *C. elegans* model.** Two types of assays were performed, i.e., therapeutic treatment and prophylactic treatment. For the liquid-based assay, a 96-well microtiter plate was filled with M9 buffer to which an overnight culture of *E. coli* OP50 (feeding bacteria) or the equivalent amount of pathogenic bacteria, with or without phage, was added (Fig. 8). Then 20 mature L4 nematodes were transferred into the solution. The total volume in the microtiter plate well was maintained at 100 $\mu$L. In order to test the robustness of the experiments, different concentrations of pathogenic bacteria were used to test the infectivity. Accordingly, overnight grown bacterial cultures were diluted to $10^3$, $10^5$, and $10^7$ CFU/mL for testing. The efficacy of phage treatment was studied using varying concentrations of bacteria and phage, i.e., 1:1, 1:10, and 1:100, which were $10^5$ CFU/mL:$10^5$ PFU/mL, $10^5$ CFU/mL:$10^6$ PFU/mL, and $10^5$ CFU/mL:$10^7$ PFU/mL.

Group 1 (control) consisted of M9 buffer (60%) with *E. coli* OP50, with $\sim 10^5$ CFU/mL (40%) and 20 nematodes. Group 2: M9 buffer, 20 nematodes, no bacteria. Group 3: M9 buffer (60%), TSB (40%), 20 nematodes. Both groups 2 and 3 were used as experimental controls. Group 4 (infection control): M9 buffer (60%), bacterial pathogen (*E. coli*, *K. pneumoniae*, or *E. cloacae*, 40%), 20 nematodes. Group 5 (heat inactivated bacteria): M9 buffer (60%), heat-inactivated bacteria (*E. coli*, *K. pneumoniae*, or *E. cloacae*, 40%), 20 nematodes. Group 6 (phage toxicity test): M9 buffer (60%), bacteriophage (*Escherichia* phage myPSH1131, *Escherichia* phage myPSH2311, *Klebsiella* phage myPSH1235, and *Enterobacter* phage

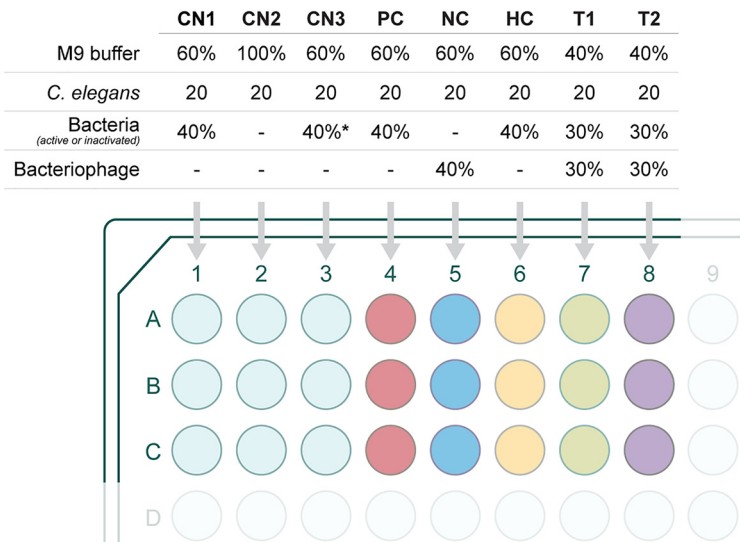

| | CN1 | CN2 | CN3 | PC | NC | HC | T1 | T2 |
|---|---|---|---|---|---|---|---|---|
| M9 buffer | 60% | 100% | 60% | 60% | 60% | 60% | 40% | 40% |
| *C. elegans* | 20 | 20 | 20 | 20 | 20 | 20 | 20 | 20 |
| Bacteria (active or inactivated) | 40% | - | 40%* | 40% | - | 40% | 30% | 30% |
| Bacteriophage | - | - | - | - | 40% | - | 30% | 30% |

**FIG 8** Representation of the study groups used to evaluate the efficacy of bacteriophages in treating bacterial infections in the *C. elegans* model. In a 96-well microtiter plate, experiments were conducted in triplicate. Details are as follows: negative controls CN1 - M9 buffer (60%), *E. coli* OP50 (40%) and 20 nematodes; CN2 - M9 buffer and 20 nematodes; CN3 - M9 buffer (60%), TSB (40%), and 20 nematodes. PC (infection control) - M9 buffer (60%), overnight bacterial culture of *E. coli*, *K. pneumoniae*, or *E. cloacae* (40%) and 20 nematodes. NC (phage control) - M9 buffer (60%), bacteriophage (*Escherichia* phage myPSH1131, *Escherichia* phage myPSH2311, *Klebsiella* phage myPSH1235, and *Enterobacter* phage myPSH1140) (40%) and 20 nematodes. HC (heat-inactivated bacteria control) - M9 buffer (60%), heat-killed bacterial culture of *E. coli* or *K. pneumoniae* or *E. cloacae* (40%) and 20 nematodes. T1 (treatment group, phages were added 2 h after bacterial infection) - M9 buffer (40%), overnight bacterial culture of *E. coli*, *K. pneumoniae*, or *E. cloacae* (30%), bacteriophage (30%), and 20 nematodes. T2 (prophylactic treatment group, phages were added 1 h before the bacterial infection) - M9 buffer (40%), bacteriophage (30%), overnight bacterial culture of *E. coli*, *K. pneumoniae*, or *E. cloacae* (30%) and 20 nematodes. *, TSB was used instead of bacteria.

myPSH1140, 40%) and 20 nematodes. Group 7 (therapeutic treatment group): M9 buffer (40%), bacterial pathogen (*E. coli*, *K. pneumoniae*, or *E. cloacae*, 30%), 20 nematodes; bacteriophage (30%) were added after 2 h of exposure to bacteria. Group 8 (prophylactic treatment group): M9 buffer (40%), bacteriophage (30%), 20 nematodes; bacterial cultures of *E. coli*, *K. pneumoniae*, or *E. cloacae* (30%) were added after 1 h. Plates were incubated at 20°C, and survival of the nematodes was monitored every 24 h for 5 days. The results were evaluated based on live nematodes (moving) and dead nematodes (lack of movement); see Fig. 1. The L4 stage nematodes were chosen for the study because even though the life span of *C. elegans* is 20 ± 3 days, nematodes drastically reduce motility when aging, making it difficult to establish if infected (47). All the experiments were repeated a minimum of three times for statistical significance. After 4 days, 10 nematodes were removed from groups 5, 6, and 7, which are phage only, therapeutic treatment, and prophylactic treatment, respectively, to determine the phage titer using the double agar overlay method. In brief, the nematodes were washed thrice with M9 buffer, vortexed, ground (with mortar and pestle), and centrifuged at $10,000 \times g$ for 5 min. The supernatant was used to determine the phage titer. All bacteriophages were used alone (not in combination, with the exception of the poly-microbial tests below), and the experiments were repeated three times for statistical significance.

The efficacy of a phage cocktail was evaluated by establishing poly-microbial infections. To this end, the bacterial cultures were mixed at equal volumes containing the same CFU numbers. Heat-inactivated bacteria were also prepared from the mixed culture. The phage cocktail was prepared by mixing the bacteriophages at equal volumes with identical PFU. The efficacy of phage treatment was studied using varying concentrations of bacteria and phage, i.e., 1:1, 1:10, and 1:100, which were $10^5$ CFU/mL:$10^5$ PFU/mL, $10^5$ CFU/mL:$10^6$ PFU/mL, and $10^5$ CFU/mL:$10^7$ PFU/mL. Group 1 consisted of M9 buffer (60%) with *E. coli* OP50 (40%) and 20 nematodes. Group 2: M9 buffer, 20 nematodes, no bacteria. Group 3: M9 buffer (60%), TSB (40%), 20 nematodes. Group 4 (infection control): M9 buffer (60%), mixed bacterial culture (*E. coli*, *K. pneumoniae*, *E. cloacae*, 40%), 20 nematodes. Group 5 (heat inactivated bacteria): M9 buffer (60%), heat-killed bacteria (poly-microbial), 20 nematodes. Group 6 (phage toxicity test): M9 buffer (60%), phage cocktail (consists of four phages, 40%), and 20 nematodes. Group 7 (therapeutic treatment group): M9 buffer (40%), mixed bacterial culture (*E. coli*, *K. pneumoniae*, *E. cloacae*, 40%), 20 nematodes; phage cocktail (30%) was added after 2 h of exposure to the bacteria. Group 8 (prophylactic treatment group): M9 buffer (40%), phage cocktail (30%), 20 nematodes; the poly-microbial culture containing *E. coli*, *K. pneumoniae*, *E. cloacae* (30%) was added after 1 h. Plates were incubated at 20°C and survival of the nematodes was monitored every 24 h for 5 days. The results were evaluated and statistical analysis was performed.

**Enumeration of bacteriophages from the treated nematodes.** To analyze the presence of bacteriophages inside the nematodes or uptake of bacteriophages by the nematodes, phage numbers were determined as follows. Briefly, after 4 days, 10 nematodes were removed from groups 5, 6, and 7, which

are phage only, therapeutic treatment, and prophylactic treatment, respectively, to determine the phage titer using the double agar overlay method. In brief, the nematodes were washed thrice with M9 buffer, vortexed, ground (with mortar and pestle), and centrifuged at 10,000 × $g$ for 5 min. The supernatant was used to determine the phage titer. All bacteriophages were used alone (not in combination), and the experiments were repeated three times for statistical significance.

**Statistical analysis.** Survival curves were plotted using the Kaplan-Meier method, and the log-rank test was used to calculate the difference in survival rates using GraphPad Prism software 7.0 (GraphPad Software, Inc., La Jolla, USA). $P < 0.05$ was considered as statistically significant (log-rank test).

## SUPPLEMENTAL MATERIAL

Supplemental material is available online only.
**SUPPLEMENTAL FILE 1**, PDF file, 0.2 MB.

## ACKNOWLEDGMENTS

We thank Vellore Institute of Technology for providing the "VIT Seed Grant," and the support provided by the Zhejiang Province Postdoctoral Research Fund (ZJ2020151) to P.M. We gratefully acknowledge the Caenorhabditis Genetic Centre (CGC, University of Minnesota, MN, USA), which is funded by the NIH Office of Research Infrastructure Programs (P40 OD010440), for providing Bristol N2 (wild type) used in this work.

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
