## [Reviewer comments · Microbiology Spectrum]

Microbiology Spectrum

A multiwell-plate *Caenorhabditis elegans* assay for assessing the therapeutic potential of Bacteriophages against Clinical Pathogens

Prasanth Manohar, Belinda Loh, Namasivayam Elangovan, Archana Loganathan, Ramesh Nachimuthu, and Sebastian Leptihn

Corresponding Author(s): Sebastian Leptihn, Edinburgh University- Zhejiang University

Review Timeline:

Submission Date:	August 27, 2021
Editorial Decision:	October 25, 2021
Revision Received:	January 1, 2022
Accepted:	January 14, 2022

Editor: David Pride

Reviewer(s): The reviewers have opted to remain anonymous.

Transaction Report:

DOI: <https://doi.org/10.1128/Spectrum.01393-21>

October 25, 2021

Prof. Sebastian Leptihn
Edinburgh University- Zhejiang University
Edinburgh-Haining
China

Re: Spectrum01393-21 (**A multiwell-plate *Caenorhabditis elegans* assay for assessing the therapeutic potential of Bacteriophages against Clinical Pathogens**)

Dear Prof. Sebastian Leptihn:

Thank you for submitting your manuscript to Microbiology Spectrum. When submitting the revised version of your paper, please provide (1) point-by-point responses to the issues raised by the reviewers as file type "Response to Reviewers," not in your cover letter, and (2) a PDF file that indicates the changes from the original submission (by highlighting or underlining the changes) as file type "Marked Up Manuscript - For Review Only". Please use this link to submit your revised manuscript - we strongly recommend that you submit your paper within the next 60 days or reach out to me. Detailed information on submitting your revised paper are below.

Link Not Available

Sincerely,

David Pride

Journals Department
Reviewer comments:

Reviewer #1 (Comments for the Author):

Manohar et al 2021 Spectrum paper

This manuscript by Manohar et al. describes a liquid-based method for evaluating phages' antibacterial performance in vivo with pathogen-infected *Caenorhabditis elegans* models. With this method, the efficacy of phage treatment is assessed based on the survival rate of *C. elegans*, by directly counting the number of live and dead *C. elegans* under a microscope. The authors tested this method with four bacterial pathogens (two *Escherichia coli* strains, one *Klebsiella pneumoniae* strain, and one *Enterobacter cloacae* strain) and their corresponding phages. For each pathogen-phage-*C. elegans* system, two types of phage treatment were evaluated - prophylactic treatment and therapeutic treatment. Additionally, the efficacy of a phage cocktail, both prophylactic and therapeutic, was assessed in poly-bacterial infections.

The main issue with this manuscript is that if this is to be a 'method paper' the method needs to be described completely. At the moment many key details are missing from Materials and Methods (see below) and the Results section does not describe the logical steps taken to create and validate the method. I would have no confidence to try this in my own lab with it as currently

written. It may be that the authors have done all this work but have just not communicated it properly.

Major Points

1. Were any controls done to ensure that whatever buffer and/or media the phage preparations were made in did not have a similar effect as a phage preparation? Unclear from text if this was or wasn't done.
 2. Description of the conditions for the therapeutic treatment group and prophylactic treatment group were only included part way through the results. At the very least the definition of the prophylactic group should have been included earlier as it left me confused and sifting through the paper when the notion was first introduced in the introduction.
 3. From the text: "The data from our assay confirms results we have previously obtained in experiments using wax worms (9)".
 - However, the reference cited [no. 9, doi: 10.5694/mja2.50355] is a review on phage therapy authored by Fabijan et al. based in Australia, and *C. elegans* infection model is not a part of that review.
 - If this was cited in error, when the proper citation is used I would like to see a bit more detailed description of how they confirm each other. Does this mean there were similar survival rates?
 4. From the Text: "... Enterobacter is known to be a part of the commensal flora in *C. elegans*."
 - So how did you account for this? Did you ensure it was not present prior to the treatment? Is it a different strain that is present to the *E. cloacae* tested?
 5. From the text "The amount of phage particles inside the worms was reduced to 102 PFU/mL from an initial concentration of 106 PFU/mL after 4 days in a solution containing only the phage (data not shown)."
 - It is not appropriate to not show data. Please add these data as supplementary information if you believe it is of a minor importance to the main aims of the paper. That being said, it seems that these are actually crucial pieces of data if the method is to be valid. They should be figures in the main text.
 - it is also very unclear how this experiment was conducted, please add additional explanation in the Results or in the figure caption.
 6. From the text: "A representative nematode scoring model is provided in Fig.6."
 - this doesn't belong as the last sentence in the Results section. If this is the foundation of the method it should probably be at the beginning of the results along with an explanation of how scoring was done (and preferably an idea of how long it takes so that we can compare to *Galleria*).
 7. The relative ratios of CFU per worm as well as PFU to CFU (Multiplicity of Infection) should be clearly described in the Results text.
 8. Immune systems of the nematode as it relates to defences against bacteria should be described and contrasted to *Galleria*. Is this a direct 'apples to apples' comparison between the species or not?
 9. The relative bacterial load and phage load within the nematode should be better described in Results.
 10. Whether phage replication is observed should be noted in the Results.
Fig 1.
 - what is the 'control' in this figure? Unclear
 - numbers of animals measured (n=?) are needed to understand the robustness of this technique and statistics
 11. To demonstrate this assay is robust, a variety of conditions should be tested. In this study, the authors used 30% bacteria and 30% phage in treating bacterial infections. As material and methods are written, the ratio of bacterial cells to phage virions is 1:10. Why was this ratio chosen? Were other ratios tested? If so, please show the data.
- Also, the sample volume for the assays is not specified; therefore, it is unknown how many bacterial cells and phage virions were in the sample.

Minor Points

Related to Fig. 1: ... phage preparations (106 PFU/mL)..
- is this concentration in the media? Unclear.

", the phage was added before the pathogen was included into the media."
- how long before?

"It is a valuable argument to state that phages....
- this isn't an argument it is a hypothesis. No observations or measurements were made of the phage replication cycle

Wax worm and wax moth are both used. If there is a good reason to use different terms for the same thing, this should be made explicit.

Reviewer #2 (Comments for the Author):

The paper by Manohar et al describes the development of a new liquid-format assay to use the nematode *C. elegans* as a model system to evaluate phage-bacteria interactions. Four bacterial pathogens are evaluated in the paper: *E. coli* (2 pathotypes), *Klebsiella pneumoniae* and *Enterobacter cloacae*. These are evaluated +/- phage treatment in mono-microbial and poly-microbial infection scenarios, and the phage treatment is contrasted in two delivery protocols: therapeutic treatment and prophylactic treatment. The paper is well written, the data is well presented, and the results support all of the conclusions drawn from the study. I see one potential (minor) weakness in the assay, but this point could be addressed in the Discussion of the paper without need of further experiments. A few minor points are made below for the authors to consider.

The assay is described in detail in order that others could benefit from its use. The only weakness in this is that the output of the assay requires an experienced operator to count worm survival. This also means that the assay is semi-quantitative rather than truly quantitative. Could these points be raised in Discussion, for example in debating whether development of an ELISA or immunoblot (dot-blot) quantitation of worms could be incorporated in a future iteration of the assay?

I appreciate that the bacterial strains have been previously described (References 39 and 40 are cited for this) but for the ease of readers who are non-experts with *E. coli*, could the two pieces of text relating to the *E. coli* pathovars be slightly embellished: "The first isolate we tested in our study was *E. coli* 131, a clinical strain from a diagnostic center in Chennai (India), isolated from blood sample and identified to be enteropathogenic (an EPEC strain)" Either cite a reference or add a sentence to be explicit about how this was determined to be EPEC.
"Next, we tested the *E. coli* strain 311 which was isolated from blood and found to be enterohemorrhagic, i.e. an EHEC strain" add a sentence to be explicit about how this was determined to be EHEC.

A strength of the paper is that two infection protocols were compared: therapeutic treatment and prophylactic treatment. It would be helpful to have a diagram, a simple time-line would be enough, to represent the difference in the two types of treatment: that is, to graphically document the order of additions in the two different treatment protocols.

Page 9: "forth", should be fourth.

Page 10: "valuable argument", should be valid argument.

Staff Comments:

Preparing Revision Guidelines

Please return the manuscript within 60 days; if you cannot complete the modification within this time period, please contact me. If you do not wish to modify the manuscript and prefer to submit it to another journal, please notify me of your decision immediately so that the manuscript may be formally withdrawn from consideration by Microbiology Spectrum.

Reviewer comments:

Reviewer #1 (Comments for the Author):

Manohar et al., 2021 Spectrum paper

This manuscript by Manohar et al. describes a liquid-based method for evaluating phages' antibacterial performance *in vivo* with pathogen-infected *Caenorhabditis elegans* models. With this method, the efficacy of phage treatment is assessed based on the survival rate of *C. elegans*, by directly counting the number of live and dead *C. elegans* under a microscope. The authors tested this method with four bacterial pathogens (two *Escherichia coli* strains, one *Klebsiella pneumoniae* strain, and one *Enterobacter cloacae* strain) and their corresponding phages. For each pathogen-phage- *C. elegans* system, two types of phage treatment were evaluated - prophylactic treatment and therapeutic treatment. Additionally, the efficacy of a phage cocktail, both prophylactic and therapeutic, was assessed in poly-bacterial infections.

The main issue with this manuscript is that if this is to be a 'method paper' the method needs to be described completely. At the moment many key details are missing from Materials and Methods (see below) and the Results section does not describe the logical steps taken to create and validate the method. I would have no confidence to try this in my own lab with it as currently written. It may be that the authors have done all this work but have just not communicated it properly.

A: We agree with the constructive comments and have extensively rewritten the method section (and others) in order to allow a group to set up the assay in the lab.

Major Points:

Q1. Were any controls done to ensure that whatever buffer and/or media the phage preparations were made in did not have a similar effect as a phage preparation? Unclear from text if this was or wasn't done.

A: Yes, the M9 buffer and TSB media used in this study were tested separately to study the toxicity. The results are presented in **S.figure 1**. Any effects potentially stemming from M9 buffer or TSB media are -if existent- neglected as the infection groups which consisted of M9 buffer and pathogenic bacteria, 100% lethality was noted, while the media or buffer alone showed no effect on the nematodes. The results are presented in **S.figure 2** and in **Figures 2,3,4,5**.

Q2. Description of the conditions for the therapeutic treatment group and prophylactic treatment group were only included part way through the results. At the very least the definition of the prophylactic group should have been included earlier as it left me confused and sifting through the paper when the notion was first introduced in the introduction.

A: Thank you for your valuable suggestion. The explanation for prophylactic (phages introduced one hour before bacterial infection) and therapeutic (phages introduced two hours after bacterial infection) is now provided at the end of the introduction.

Q3. From the text: "The data from our assay confirms results we have previously obtained in experiments using wax worms (9)".

- However, the reference cited [no. 9, doi: 10.5694/mja2.50355] is a review on phage therapy authored by Fabijan et al. based in Australia, and C. elegans infection model is not a part of that review.

- If this was cited in error, when the proper citation is used I would like to see a bit more detailed description of how they confirm each other. Does this mean there were similar survival rates?

A: Thanks for identifying our mistake. The reference was indeed cited wrongly, which we now corrected. The following sentence was added to describe the previous study, "In our previous study, when wax worms were infected with bacteria (10^8 CFU/mL) and treated with phages (10^4 PFU/mL), up to 80% larval survival was noted".

Q4. From the Text: "... Enterobacter is known to be a part of the commensal flora in C. elegans."

- So how did you account for this? Did you ensure it was not present prior to the treatment? Is it a different strain that is present to the E. cloacae tested?

A: *Enterobacter* species can be part of the commensal flora in *C. Elegans* [Berg M, Zhou XY, Shapira M. Host-specific functional significance of *Caenorhabditis gut commensals*. *Frontiers in microbiology*. 2016 Oct 17;7:1622]. In this study, the pathogenic *E. cloacae* strain 140 was isolated from humans. The *Enterobacter* phage myPSH1140 was able to infect this human *E. cloacae* isolate. To eliminate the possibility of potential cross sensitivity of our *Enterobacter* phage myPSH1140 against the commensal *Enterobacter* in the *C. elegans*, we attempted to isolate *Enterobacter* from our worms using selective media, metabolic tests and 16S sequencing. In our hands, our *C. elegans* did not contain any *Enterobacter sp.* We have added this information in our manuscript. In addition, we observed no toxic/ adverse effects when giving the *Enterobacter* phage myPSH1140 alone (**Figure 5**).

Q5. From the text "The amount of phage particles inside the worms was reduced to 102 PFU/mL from an initial concentration of 10⁶ PFU/mL after 4 days in a solution containing only the phage (data not shown)."

- It is not appropriate to not show data. Please add these data as supplementary information if you believe it is of a minor importance to the main aims of the paper. That being said, it seems that these are actually crucial pieces of data if the method is to be valid. They should be figures in the main text.

- it is also very unclear how this experiment was conducted, please add additional explanation in the Results or in the figure caption.

A: We agree with the reviewer's suggestion and now show the data in our manuscript. The results are represented in **Figure 7**.

Q6. From the text: "A representative nematode scoring model is provided in Fig.6."

- this doesn't belong as the last sentence in the Results section. If this is the foundation of the method it should probably be at the beginning of the results along with an explanation of how scoring was done (and preferably an idea of how long it takes so that we can compare to *Galleria*).

A: Thank you for the suggestion. Figure 6 has now been changed to **figure 1** and the following sentence was added to the beginning of the results section, "Nematodes were evaluated based on appearance (reflectance and optical transparency) and mobility (alive: moving; dead: inactive) as observed under an inverted microscope (40x magnification). A representative scoring model is provided in Fig.1. Scoring was performed in 24 hour intervals."

Q7. The relative ratios of CFU per worm as well as PFU to CFU (Multiplicity of Infection) should be clearly described in the Results text.

A: Thanks for the suggestion. The results section has been modified with the additional data.

Q8. Immune systems of the nematode as it relates to defenses against bacteria should be described and contrasted to *Galleria*. Is this a direct 'apples to apples' comparison between the species or not?

A: This is indeed a very difficult question to answer. As a recent review describes, innate immune factors are conserved among nematodes, insects, and mammals which allowed identifying "important immune factors in *C. elegans*, indicating similarities between innate immunity in *C. elegans* and other metazoa." [Leah J. Radeke and Michael A. Herman, Microbiology and Molecular Biology Reviews, 85, 2, e00146-20, 2021, doi: 10.1128/MMBR.00146-20]. Both, *Galleria* and *Caenorhabditis* lack an

adaptive immune system; however research on the innate immune systems of insects and nematodes has indicated that there are differences in pattern- recognition- molecules and signaling factors. It is a very valuable suggestion to include a short note on this complex topic as nematodes and insect larvae might potentially show different responses to bacteria when used as a model test system. In our manuscript we now include this aspect in the discussion.

Q9. The relative bacterial load and phage load within the nematode should be better described in Results.

A: We agree with the reviewer and now describe the results, as represented in **figure 7**.

Q10. Whether phage replication is observed should be noted in the Results.

Fig 1.

-what is the 'control' in this figure? Unclear

- numbers of animals measured (n=?) are needed to understand the robustness of this technique and statistics.

A: Thank you for the comments. Based on the differences that were observed in the phage enumeration experiments with and without bacteria, we can conclude that phage replication happens inside the nematodes (**figure 7**).

Figures 2,3,4,5,6 has been corrected to include the statement on control and number of animals used.

Q11. To demonstrate this assay is robust, a variety of conditions should be tested. In this study, the authors used 30% bacteria and 30% phage in treating bacterial infections. As material and methods are written, the ratio of bacterial cells to phage virions is 1:10. Why was this ratio chosen? Were other ratios tested? If so, please show the data.

Also, the sample volume for the assays is not specified; therefore, it is unknown how many bacterial cells and phage virions were in the sample.

A: Thank you for the suggestion. Additional experiments were performed to show the robustness of this assay or study. Accordingly, the efficacy of phage treatment was studied using varying concentrations of bacteria and phage i.e. 1:1, 1:10 and 1:100 which were 10^5 CFU/mL: 10^5 PFU/mL, 10^5 CFU/mL: 10^6 PFU/mL and 10^5 CFU/mL: 10^7 PFU/mL.

12. There is not enough information for others to repeat the experiments. Instruments and some experimental parameters are not specified. Here to list a few:

1) What was the sample volume per well on a 96-well microplate?

A: The sample volume was always maintained at 100 microliters. A sentence has been added in the text.

2) Growth condition of bacterial strains - how was it shaken?

A: For the assay, bacterial strains were grown overnight at 37°C with shaking at 120 rpm, then the culture was diluted to $OD_{600}=0.6$ ($\sim 10^5$ CFU/mL) or required concentration. A sentence has been added in the text.

3) Where were the materials (C. elegans, bacterial strains, phages, reagents) obtained/ purchased from?

A: Thank you for the suggestion. All the information has been added in the text.

Minor Points:

Related to Fig. 1: ... phage preparations (106 PFU/mL)

- is this concentration in the media? Unclear

A: It has been corrected.

" , the phage was added before the pathogen was included into the media."

- how long before?

A: Phage was added one hour before the bacteria. It has been included.

"It is a valuable argument to state that phages...."

- this isn't an argument it is a hypothesis. No observations or measurements were made of the phage replication cycle.

A: This has been corrected.

Wax worm and wax moth are both used. If there is a good reason to use different terms for the same thing, this should be made explicit.

A: Wax moth has been removed and replaced by "wax-worm" consistently.

Reviewer #2 (Comments for the Author):

The paper by Manohar et al describes the development of a new liquid-format assay to use the nematode C. elegans as a model system to evaluate phage-

bacteria interactions. Four bacterial pathogens are evaluated in the paper: *E. coli* (2 pathotypes), *Klebsiella pneumoniae* and *Enterobacter cloacae*. These are evaluated +/- phage treatment in mono-microbial and poly-microbial infection scenarios, and the phage treatment is contrasted in two delivery protocols: therapeutic treatment and prophylactic treatment. The paper is well written, the data is well presented, and the results support all of the conclusions drawn from the study. I see one potential (minor) weakness in the assay, but this point could be addressed in the Discussion of the paper without need of further experiments. A few minor points are made below for the authors to consider.

Q1. The assay is described in detail in order that others could benefit from its use. The only weakness in this is that the output of the assay requires an experienced operator to count worm survival. This also means that the assay is semi-quantitative rather than truly quantitative. Could these points be raised in Discussion, for example in debating whether development of an ELISA or immunoblot (dot-blot) quantitation of worms could be incorporated in a future iteration of the assay?

A: Thank you for the suggestion. Suggestions for quantitative (observer-independent) approaches have now been included in the discussion.

Q2. I appreciate that the bacterial strains have been previously described (References 39 and 40 are cited for this) but for the ease of readers who are non-experts with *E. coli*, could the two pieces of text relating to the *E. coli* pathovars be slightly embellished:

"The first isolate we tested in our study was *E. coli* 131, a clinical strain from a diagnostic center in Chennai (India), isolated from blood sample and identified to be enteropathogenic (an EPEC strain)"Either cite a reference or add a sentence to be explicit about how this was determined to be EPEC. "Next, we tested the *E. coli* strain 311 which was isolated from blood and found to be enterohemorrhagic, i.e. an EHEC strain"add a sentence to be explicit about how this was determined to be EHEC.

A: Thank you for the suggestion. The identification of *E. coli* pathotypes was done using polymerase chain reaction as explained in our previous study. This has now been included in the materials and methods and results section. References 39 and 40 have also been included at the end of the sentence.

Q3: A strength of the paper is that two infection protocols were compared: therapeutic treatment and prophylactic treatment. It would be helpful to have a diagram, a simple time-line would be enough, to represent the difference in the

two types of treatment: that is, to graphically document the order of additions in the two different treatment protocols.

A: Thank you for this idea which will indeed be very valuable for the reader. We have now included such a diagram in the manuscript (Figure 1) as an overview of the entire process serving as a visual guide for the assay. The schematic also includes when the prophylactic phages are added, or when the therapeutic treatment was started, as well as the optical scoring system for the nematodes.

Page 9: "forth", should be fourth.

A: Corrected.

Page 10: "valuable argument", should be valid argument.

A: Corrected.

January 14, 2022

Prof. Sebastian Leptihn
Edinburgh University- Zhejiang University
Edinburgh-Haining
China

Re: Spectrum01393-21R1 (**A multiwell-plate *Caenorhabditis elegans* assay for assessing the therapeutic potential of Bacteriophages against Clinical Pathogens**)

Dear Prof. Sebastian Leptihn:

Your manuscript has been accepted, and I am forwarding it to the ASM Journals Department for publication. You will be notified when your proofs are ready to be viewed.

Sincerely,

David Pride
Editor, Microbiology Spectrum
